# Prony Method for Two-Generator Sparse Expansion Problem

**Abdulmtalb Hussen** [1,*] and **Wenjie He** [2]

1 School of Engineering, Math and Technology Navajo Technical University, Lowerpoint Rd State Hwy 371, Crownpoint, NM 87313, USA
2 Department of Computer Science, University of Missouri, St. Louis, MO 63121, USA; hew@umsl.edu
* Correspondence: ahussen@navajotech.edu

**Abstract:** In data analysis and signal processing, the recovery of structured functions from the given sampling values is a fundamental problem. Many methods generalized from the Prony method have been developed to solve this problem; however, the current research mainly deals with the functions represented in sparse expansions using a *single generating function*. In this paper, we generalize the Prony method to solve the sparse expansion problem for *two generating functions*, so that more types of functions can be recovered by Prony-type methods. The two-generator sparse expansion problem has some special properties. For example, the two sets of frequencies need to be separated from the zeros of the Prony polynomial. We propose a *two-stage least-square detection method* to solve this problem effectively.

**Keywords:** Prony method; exponential sums; eigenfunctions; eigenvalues; sparse expansion; generating function; Hankel matrix; short time Fourier transform; least-square method

## 1. Introduction

The Prony method is a popular tool used to recover the functions represented in sparse expansions using one generating function. For example, the function with the following form

$$f(x) = \sum_{j=1}^{M} c_j e^{ix\phi_j} \tag{1}$$

can be recovered from $2M$ equispaced sampling values $f(lh), l = 0, \ldots, 2M - 1$ for an appropriate positive constant $h$; however, in many real-world applications, we need to deal with the functions represented by more than one generating functions. For example, the *harmonic signals* with the form

$$f(x) = \sum_{j=1}^{M} \big( c_j \cos(\phi_j x) + d_j \sin(\beta_j x) \big), \tag{2}$$

are generated by two generating functions (or simply generators): $\cos(\phi x)$ and $\sin(\beta x)$, where $\phi$ and $\beta$ are generic parameters used as the placeholders for the real parameters $\{\phi_j\}_{j=1}^{M}$ and $\{\beta_j\}_{j=1}^{M}$ to generate the specific terms in the expansion. In this system, we have two sets of coefficients $\{c_j\}_{j=1}^{M}$ and $\{d_j\}_{j=1}^{M}$ and two sets of frequencies $\{\phi_j\}_{j=1}^{M}$ and $\{\beta_j\}_{j=1}^{M}$. Analogous to the original Prony method, we expect to use $4M$ equispaced sampling values $f(lh), l = 0, \ldots, 4M - 1$ to recover those four sets of parameters.

There are some existing methods to solve this problem. The first one is to convert it to a single-generator problem by the following formulas

$$\cos x = \frac{1}{2}(e^{ix} + e^{-ix}) \quad \text{and} \quad \sin x = \frac{1}{2i}(e^{ix} - e^{-ix}),$$

which results in problem (1) (see [1]). Another way using the same idea is based on the *even/odd* properties for $\cos x$ and $\sin x$ (see [2]) as follows

$$f(x) + f(-x) = 2\sum_{j=1}^{M} c_j \cos(\phi_j x). \tag{3}$$

However, this approach is very restrictive, because the chance that one can make this kind of conversion is very small. In this paper, we are interested in solving the *general* two-generator sparse expansion problem by a new way of generalized Prony method. More specifically, we study the functions with the following two-generator sparse expansion

$$f(x) = \sum_{j=1}^{M_1} c_j u(\phi_j x) + \sum_{l=1}^{M_2} d_l v(\beta_l x), \tag{4}$$

where $u(\phi x)$ and $v(\beta x)$ are two different functions used as the generators. In order to make the Prony method work, we need a critical condition for our special technique: *There exists a linear operator, such that $u(\phi x)$ and $v(\beta x)$ are both eigenfunctions of this operator.*

Another situation that could lead to the two-generator expansion problem is when we apply some special transforms on a sparse expansion. For example, when we apply the *short time Fourier transform* (STFT), i.e.,

$$\text{STFT}\{f(x)\}(\omega, \tau) = \int_{-\infty}^{\infty} f(x) w(x - \tau) e^{-i\omega x} dx \tag{5}$$

using the Gaussian window function $w(x) = \frac{1}{\sqrt{2\pi}} e^{-\frac{x^2}{2\sigma^2}}$ on the sparse cosine expansion

$$f(x) = \sum_{j=1}^{M} c_j \cos(\phi_j x), \tag{6}$$

we would obtain a two-generator sparse expansion as follows,

$$f(x) = \sum_{j=1}^{M} c_j e^{-\beta(\phi_j - x)^2} + \sum_{j=1}^{M} c_j e^{-\beta(\phi_j + x)^2}. \tag{7}$$

In this example, the two generators are $e^{-\beta(\phi - x)^2}$ and $e^{-\beta(\phi + x)^2}$ with $\beta \neq 0$. Actually, the original single-generator problem (6) can be solved directly. For example, one can convert $\cos(\phi x)$ to $\frac{1}{2}(e^{i\phi x} + e^{-i\phi x})$ (see [1]), or use a method based on the Chebyshev polynomials (see [3]). When we solve problem (6) directly, we use the sampling values in the time domain; when we solve the problem in the form of (7), we use the sampling values in the frequency domain. (See [4] for a discussion on sampling values in the frequency domain.) In this paper, we use this example to study the special properties of the two-generator sparse expansion problem.

Since the signals could take various forms, not necessarily in the exponential form studied in the classical Prony method, many researchers generalized the Prony method to handle different types of signals. For example, many results in [1,3,5–12] have been developed over the last few years. In particular, Peter and Plonka in [1,8] generalized the Prony method to reconstruct M-sparse expansions in terms of eigenfunctions of some special linear operators. In [3], Plonka and others reconstructed different signals by exploiting the generalized shift operator. These results provide us the building blocks for our method in this paper.

We organize our presentation in the remaining sections as follows. In Section 2, we quickly review the classical Prony method and one of its generalizations for the Gaussian generating function to establish the foundation of our method. In Section 3, we describe the

details of our method using the example with two generators: cosine and sine functions. In Section 4, we apply our method on two different types of Gaussian generating functions, so that we can study an interesting property: *When the Hankel matrix for finding the coefficients of the Prony polynomial is singular, what does it really mean?* In Section 5, we show two examples that correspond to the two problems solved in Sections 3 and 4, respectively. Finally, we make conclusions in Section 6 and describe two related research problems to be solved in the future.

## 2. Review of the Prony Method and One of Its Generalizations

Our method is built on top of the Prony method and one of its generalizations. Before we present our technique, we review these basic methods.

### 2.1. Classical Prony Method

Let $f(x)$ be a function in the form of

$$f(x) = \sum_{j=1}^{M} c_j e^{-ix\phi_j} \tag{8}$$

with $M \geq 1$. Then the coefficients $\{c_j\}_1^M$ and the frequencies $\{\phi_j\}_1^M$ can be recovered from the sampling values $f(lh), l = 0, ..., 2M - 1$, where $h$ is some positive constant. To solve this problem, a special polynomial called the Prony polynomial can help us convert the relatively hard *non-linear* problem (8) to two *linear* problems and a *simple non-linear* problem (finding zeros of a polynomial). The Prony polynomial for (8) is defined as

$$\Lambda(z) = \prod_{j=1}^{M} (z - e^{-ih\phi_j}) = \sum_{l=0}^{M} \lambda_l z^l, \tag{9}$$

where $\lambda_l, l = 0, ..., M$ are the coefficients of the monomial terms in (9) with the leading coefficient $\lambda_M = 1$. The technique is based on the following critical property:

$$\sum_{l=0}^{M} \lambda_l f(h(l+m)) = \sum_{l=0}^{M} \lambda_l \sum_{j=1}^{M} c_j e^{-ih(l+m)\phi_j} = \sum_{j=1}^{M} c_j e^{-ihm\phi_j} \underbrace{\sum_{l=0}^{M} \lambda_l e^{-ihl\phi_j}}_{=0} = 0 \tag{10}$$

for any $m = 0, 1, \ldots, M - 1$, which can be written as the following linear system

$$\left[ f(h(l+m)) \right]_{l,m=0}^{M-1} \begin{bmatrix} \lambda_0 \\ \vdots \\ \lambda_{M-1} \end{bmatrix} = - \begin{bmatrix} f(hM) \\ \vdots \\ f(h(2M-1)) \end{bmatrix}. \tag{11}$$

The coefficient vector $\boldsymbol{\lambda} = [\lambda_0, \lambda_1, \ldots, \lambda_{M-1}]^T$ can be calculated from the $2M$ sampling values $f(lh), l = 0, ..., 2M - 1$. The linear system (11) is guaranteed to have a unique solution under the condition that all $\phi_j$'s are distinct in $(-K, K) \subset \mathbb{R}$ for some $K > 0$ (with $h$ in the range $0 < h < \frac{\pi}{K}$), and $c_1, \ldots, c_M$ are nonzero in $\mathbb{C}$, which is a natural requirement for problem (8). This property is a direct result of the following matrix factorization

$$\left[ f(h(l+m)) \right]_{l,m=0}^{M-1} = \boldsymbol{V}^T \text{diag}(c_1, ..., c_M) \boldsymbol{V}, \tag{12}$$

where $V := [e^{-ilh\phi_j}]_{l=0,j=1}^{l=M-1,j=M}$ is a Vandermonde matrix, which is non-singular for distinct $\phi_j$'s and $h\phi_j \in (-\pi, \pi]$ for $j = 1, ..., M$. The frequencies can be extracted from the zeros of $\Lambda(z)$ (in the form of $z_j = e^{-ih\phi_j}$) using the formula

$$\phi_j = \frac{-\text{Im}(\ln(z_j))}{h}, \quad j = 1, ..., M. \tag{13}$$

Finally, the coefficients $c_j, j = 1, ..., M$ can be determined by solving the following *overdetermined* linear system (with $M$ unknowns and $2M$ equations)

$$f(lh) = \sum_{j=1}^{M} c_j e^{-ilh\phi_j}, \quad l = 0, ..., 2M - 1. \tag{14}$$

The redundant equations in this overdetermined linear system will play a critical role in our two-generator method to help us separate the frequencies associated with the two generators (see Section 3).

### 2.2. Sparse Expansions on Shifted Gaussian

In order to solve the two-generator sparse expansion problem (7), we need to apply the technique presented in [3], which solves a single-generator sparse expansion problem with the following form

$$f(x) = \sum_{j=1}^{M} c_j e^{-\beta(x-\phi_j)^2}, \tag{15}$$

where $\beta \in \mathbb{C}\backslash\{0\}$. The technique relies on the following generalized shift operator

$$\mathcal{S}_{K,h} f(x) = K(x,h) f(x+h), \tag{16}$$

where $h \neq 0$, and $K(\cdot, \cdot)$ has the property

$$K(x, h_1 + h_2) = K(x, h_1)K(x + h_1, h_2) = K(x, h_2)K(x + h_2, h_1).$$

The $K(x, h)$ function in (16) is chosen to be $e^{\beta h(2x+h)}$, so that we have the following critical property

$$(\mathcal{S}_{K,h} e^{-\beta(\phi-\cdot)^2})(x) = e^{2\beta\phi h} e^{-\beta(\phi-x)^2}, \tag{17}$$

which means that $e^{-\beta(\phi_j - x)}$'s are eigenfunctions of $\mathcal{S}_{K,h}$ for all $\phi_j \in \mathbb{R}$.

The sparse expansion $f(x)$ in (15) can be reconstructed using $2M$ sampling values $f(x_0 + hk), k = 0, ..., 2M - 1$, and $x_0$ is an arbitrary real number. If $\text{Re}\,\beta \neq 0$, then $h \in \mathbb{R}\backslash\{0\}$; while if $\text{Re}\,\beta = 0$, then $0 < h \leq \frac{\pi}{2|\text{Im}\beta|L}$ with $\phi_j \in (-L, L)$ for $j = 1, ..., M$ for some given $L$. (See [3].) The Prony polynomial for the problem in (15) can be defined as:

$$\Lambda(z) := \prod_{j=1}^{M}(z - e^{2h\beta\phi_j}) = \sum_{l=0}^{M} \lambda_l z^l \tag{18}$$

with $\lambda_M = 1$. Then, we have the following linear system

$$\sum_{l=0}^{M-1} \lambda_l e^{\beta h(l+m)(2x_0+h(l+m))} f(x_0 + h(l+m)) = -e^{\beta h(m+M)(2x_0+h(m+M))} f(x_0 + h(m+M)) \tag{19}$$

for $m = 0, 1, ..., M - 1$, which can be represented as an inhomogeneous system

$$H\lambda = -G, \tag{20}$$

where $G := \left[(\mathcal{S}_{K,(M+m)h}f)(x_0)\right]_{m=0}^{M-1}$, and $H := \left[(\mathcal{S}_{K,(l+m)h}f)(x_0 + (l+m)h)\right]_{l,m=0}^{M-1}$. This $H$ matrix is a Hankel matrix, and it has the following structure

$$
H := \left[(\mathcal{S}_{K,(l+m)h}f)(x_0 + (l+m)h)\right]_{l,m=0}^{M-1} = \left[K(x_0, (l+m)h)f(x_0 + (l+m)h)\right]_{l,m=0}^{M-1} \quad (21)
$$
$$
= V\mathrm{diag}(c_j e^{-\beta(\phi_j - x_0)^2})V^T,
$$

with the Vandermonde matrix

$$
V := \begin{bmatrix}
1 & 1 & \dots & 1 \\
e^{2\beta h\phi_1} & e^{2\beta h\phi_2} & \dots & e^{2\beta h\phi_M} \\
\vdots & \vdots & \dots & \vdots \\
e^{2(M-1)\beta h\phi_1} & e^{2(M-1)\beta h\phi_2} & \dots & e^{2(M-1)\beta h\phi_M}
\end{bmatrix}.
$$

Thus, $H$ is invertible for distinct $\phi_j$'s in $(-L, L) \subset \mathbb{R}$ for $L > 0$, and the vector of the coefficients $\lambda := [\lambda_0, ..., \lambda_{M-1}]^T$ are obtained by solving the system (20), which can be used to calculate the parameters $\{\phi_j\}$'s.

Finally, the coefficients $c_j$'s in the expansion (15) can be computed by solving the following overdetermined linear system:

$$
f(x_0 + lh) = \sum_{j=1}^{M} c_j e^{-\beta(x_0 - \phi_j + lh)^2}, \quad l = 0, ..., 2M - 1. \quad (22)
$$

### 3. The Sparse Expansion Problem with Two Generators: Cosine and Sine

In this section, we present our method for solving the two-generator sparse expansion problem in the following form

$$
f(x) = \sum_{j=1}^{M_1} c_j \cos(\phi_j x) + \sum_{l=1}^{M_2} d_l \sin(\beta_l x) \quad (23)
$$

through a modified Prony method. We present our method in the following theorem.

**Theorem 1.** *Assume that a function $f(x)$ has the two-generator sparse expansion form of* (23), *where the number of terms for two generators $M_1$ and $M_2$ are known, but the two sets of coefficients in $\{c_1, \dots, c_{M_1}\}$ and $\{d_1, \dots, d_{M_2}\}$ and the two sets of frequencies in $\{\phi_1, \dots, \phi_{M_1}\}$ and $\{\beta_1, \dots, \beta_{M_2}\}$ are unknown. If $4(M_1 + M_2) - 1$ equispaced sampling values of the form $f(x_0 + kh)$ for $k = -2(M_1 + M_2) + 1, \dots, -1, 0, 1, \dots, 2(M_1 + M_2) - 1$ are provided, then the original function $f(x)$ can be uniquely reconstructed under the following conditions:*

$1°$ *All the coefficients $\{c_1, \dots, c_{M_1}, d_1, \dots, d_{M_2}\}$ are nonzero in $\mathbb{C}$.*

$2°$ *All the frequencies $\{\phi_1, \dots, \phi_{M_1}, \beta_1, \dots, \beta_{M_2}\}$ are distinct in a range $[0, K) \subset \mathbb{R}$ for some $K > 0$. Furthermore, h is selected from the range $0 < h < \dfrac{\pi}{K}$.*

$3°$ *The value of $x_0 \in \mathbb{R}$ is selected to make the $(M_1 + M_2)$ numbers $\cos(\phi_1 x_0), \dots, \cos(\phi_{M_1} x_0)$, $\sin(\beta_1 x_0), \dots, \sin(\beta_{M_2} x_0)$ nonzero.*

**Proof.** First, we choose an appropriate linear operator, such that our two generating functions $\cos(\phi x)$ and $\sin(\beta x)$ in (23) are both the eigenfunctions of this operator. We consider the *symmetric shift operator* (see [3])

$$
\mathcal{S}_{h,-h}f(x) := \left(\frac{\mathcal{S}_{-h} + \mathcal{S}_h}{2}\right)f(x) = \frac{f(x-h) + f(x+h)}{2}. \quad (24)
$$

When we apply this operator on $\cos(\phi x)$ and $\sin(\beta x)$, we obtain

$$
\begin{aligned}
(\mathcal{S}_{h,-h})\cos(\phi x) &= \cos(\phi h)\cos(\phi x), \\
(\mathcal{S}_{h,-h})\sin(\beta x) &= \cos(\beta h)\sin(\beta x),
\end{aligned}
\tag{25}
$$

where $\cos(\phi h)$ and $\cos(\beta h)$ are the eigenvalues. Now we define the Prony polynomial for problem (23) using all the eigenvalues $\{\cos(\phi_j h)\}_{j=1}^{M_1}$ and $\{\cos(\beta_l h)\}_{l=1}^{M_2}$ as follows:

$$
\Lambda(z) = \prod_{j=1}^{M_1}(z - \cos(h\phi_j))\prod_{l=1}^{M_2}(z - \cos(h\beta_l)),
\tag{26}
$$

which can be written in terms of the Chebyshev polynomials as

$$
\Lambda(z) = \sum_{k=0}^{M_1+M_2}\lambda_k T_k(z),
\tag{27}
$$

where $T_k(z) := \cos(k\cos^{-1}(z))$. (See [3] for more information on this technique.) Since the leading coefficient of the Chebyshev polynomial $T_k(z)$ is $2^{k-1}$, we choose $\lambda_{M_1+M_2} = 2^{1-(M_1+M_2)}$, so that $\Lambda(z)$ in (27) has the leading coefficient 1. This Prony polynomial has the following critical property:

$$
\sum_{k=0}^{M_1+M_2}\lambda_k T_k(\cos(\phi_j h)) = 0 \quad \text{and} \quad \sum_{k=0}^{M_1+M_2}\lambda_k T_k(\cos(\beta_l h)) = 0
$$

for $j = 1, 2, \ldots, M_1$ and $l = 1, 2, \ldots, M_2$, respectively, which is essential to help us derive the following linear system.

To derive a linear system for $\{\lambda_k\}_{k=0}^{M_1+M_2-1}$, we need to calculate the following expression

$$
\sum_{k=0}^{M_1+M_2}\lambda_k\left(\mathcal{S}_{kh,-kh}\mathcal{S}_{mh,-mh}f(x_0)\right),
$$

which can be shown to be zero. That is,

$$
\frac{1}{4}\sum_{k=0}^{M_1+M_2}\lambda_k\left(f(x_0+(m+k)h)+f(x_0-(m+k)h)+f(x_0+(m-k)h)+f(x_0-(m-k)h)\right) = 0
\tag{28}
$$

for $m = 0, 1, \ldots, M_1 + M_2 - 1$. Indeed, using the right-hand-side expression in (23) for $f(x)$ in (28) and for a fixed $m \in \{0, 1, \ldots, M_1 + M_2 - 1\}$, we obtaing

$$
\begin{aligned}
&\frac{1}{4}\sum_{k=0}^{M_1+M_2}\lambda_k\left[\sum_{j=1}^{M_1}2c_j\left(\cos(\phi_j(x_0+mh))+\cos(\phi_j(x_0-mh))\right)\cos(\phi_j kh)\right] \\
&+\frac{1}{4}\sum_{k=0}^{M_1+M_2}\lambda_k\left[\sum_{l=1}^{M_2}2d_l\left(\sin(\beta_l(x_0+mh))+\sin(\beta_l(x_0-mh))\right)\cos(\beta_l kh)\right] \\
&= \sum_{j=1}^{M_1}c_j\cos(\phi_j x_0)\cos(\phi_j mh)\left(\sum_{k=0}^{M_1+M_2}\lambda_k\cos(\phi_j kh)\right)+\sum_{l=1}^{M_2}d_l\sin(\beta_l x_0)\cos(\beta_l mh)\left(\sum_{k=0}^{M_1+M_2}\lambda_k\cos\beta_l(kh)\right) \\
&= \sum_{j=1}^{M_1}c_j\cos(\phi_j x_0)\cos(\phi_j mh)\left(\underbrace{\sum_{k=0}^{M_1+M_2}\lambda_k T_k(\cos(\phi_j h))}_{=0}\right)+\sum_{l=1}^{M_2}d_l\sin(\beta_l x_0)\cos(\beta_l mh)\left(\underbrace{\sum_{k=0}^{M_1+M_2}\lambda_k T_k(\cos(\beta_l h))}_{=0}\right) \\
&= 0.
\end{aligned}
$$

We can reformulate the system (28) as

$$\sum_{k=0}^{(M_1+M_2)-1} \lambda_k \Big( f(x_0 + (m+k)h) + f(x_0 - (m+k)h) + f(x_0 + (m-k)h) + f(x_0 - (m-k)h) \Big)$$

$$= -2^{1-(M_1+M_2)} \Big( f(x_0 + ((M_1+M_2)+m)h) + f(x_0 - ((M_1+M_2)+m)h) \tag{29}$$

$$+ f(x_0 + ((M_1+M_2)-m)h) + f(x_0 - ((M_1+M_2)-m)h) \Big)$$

for $m = 0, 1, \ldots, M_1 + M_2 - 1$. To solve this system, we need $4(M_1 + M_2) - 1$ sampling values in the form of $f(x_0 + kh)$ for $k = -2(M_1 + M_2) + 1, \ldots, -1, 0, 1, \ldots, 2(M_1 + M_2) - 1$.

In order to see that the linear system in (29) has a unique solution, we study the $(M_1 + M_2) \times (M_1 + M_2)$ coefficient matrix in (29), which we denote as $\boldsymbol{H}$. As in the classical Prony method, we can factorize $\boldsymbol{H}$ in the following structure

$$\boldsymbol{H} := \Big[ f(x_0 + (m+k)h) + f(x_0 - (m+k)h) + f(x_0 + (m-k)h) + f(x_0 - (m-k)h) \Big]_{m,k=0}^{(M_1+M_2)-1}$$

$$= 4 \Big[ \sum_{j=1}^{M_1} c_j \cos(\phi_j x_0) \cos(\phi_j mh) \cos(\phi_j kh) + \sum_{l=1}^{M_2} d_l \sin(\beta_l x_0) \cos(\beta_l mh) \cos(\beta_l kh) \Big]_{m,k=0}^{(M_1+M_2)-1}$$

$$= 4 \boldsymbol{V}_h \boldsymbol{D} \boldsymbol{V}_h^T,$$

where the Vandermonde Block matrix $\boldsymbol{V}_h$ can be written as

$$\boldsymbol{V}_h := \begin{bmatrix} \boldsymbol{A} & | & \boldsymbol{B} \end{bmatrix}, \tag{30}$$

with

$$\boldsymbol{A} := \begin{bmatrix} 1 & \cdots & 1 \\ T_1(\cos \phi_1 h) & \cdots & T_1(\cos \phi_{M_1} h) \\ \vdots & \cdots & \vdots \\ T_{(M_1+M_2)-1}(\cos \phi_1 h) & \cdots & T_{(M_1+M_2)-1}(\cos \phi_{M_1} h) \end{bmatrix}_{(M_1+M_2) \times M_1} \tag{31}$$

and

$$\boldsymbol{B} := \begin{bmatrix} 1 & \cdots & 1 \\ T_1(\cos \beta_1 h) & \cdots & T_1(\cos \beta_{M_2} h) \\ \vdots & \cdots & \vdots \\ T_{(M_1+M_2)-1}(\cos \beta_1 h) & \cdots & T_{(M_1+M_2)-1}(\cos \beta_{M_2} h) \end{bmatrix}_{(M_1+M_2) \times M_2}, \tag{32}$$

and the diagonal block matrix $\boldsymbol{D}$ can be written as

$$\boldsymbol{D} := \left[ \begin{array}{c|c} \boldsymbol{D1} & \boldsymbol{0} \\ \hline \boldsymbol{0} & \boldsymbol{D2} \end{array} \right] \tag{33}$$

where

$$\boldsymbol{D1} := \begin{bmatrix} c_1 \cos(\phi_1 x_0) & & \\ & \ddots & \\ & & c_{M_1} \cos(\phi_{M_1} x_0) \end{bmatrix} \tag{34}$$

and

$$\boldsymbol{D2} := \begin{bmatrix} d_1 \sin(\beta_1 x_0) & & \\ & \ddots & \\ & & d_{M_2} \sin(\beta_{M_2} x_0) \end{bmatrix}. \tag{35}$$

Thus, $H$ is guaranteed to be invertible by the conditions 2° and 3° of the theorem. Then, we can find the unique solution for $\{\lambda_k\}_{k=0}^{M_1+M_2-1}$ from the linear system (29).

With these $\lambda_k$ values for $\Lambda(z)$ as in (26), we can determine $\phi_j$'s and $\beta_l$'s from the zeros of $\Lambda(z)$; however, this step is non-trivial, because we do not know what zeros correspond to $\phi_j$'s and what zeros correspond to $\beta_l$'s. In order to resolve this ambiguity, we consider all the possible cases: Among $M_1 + M_2$ zeros of $\Lambda(z)$, $M_1$ of them correspond to $\phi_j$'s. Thus, there are a total $\binom{M_1+M_2}{M_1}$ possible choices for $\phi_j$'s, among which there is exactly one choice for the solution; however, how do we select the right one? We need to go to the next *overdetermined* linear system for the answer.

When we determine the coefficients $c_j$'s and $d_l$'s in (23), we have the following linear system

$$f(x_0 + hn) = \sum_{j=1}^{M_1} c_j \cos(\phi_j(x_0 + hn)) + \sum_{l=1}^{M_2} d_l \sin(\beta_l(x_0 + hn)) \tag{36}$$

for $n = -2(M_1 + M_2) + 1, \ldots, -1, 0, \ldots, 2(M_1 + M_2) - 1$ corresponding to all the sampling values, which has $4(M_1 + M_2) - 1$ equations and $M_1 + M_2$ unknowns. This *overdetermined* linear system gives us the extra information we need to select the true-solution case from the remaining non-solution cases.

Our method is based on an observation: The sampling values $\{f(x_0 + nh)\}_{n=-2(M_1+M_2)+1}^{2(M_1+M_2)-1}$ are calculated using the original $\phi_j$'s and $\beta_l$'s (corresponding to the true-solution case), which means that all the $4(M_1 + M_2) - 1$ equations in (36) are completely satisfied for the true-solution case. In other words, the least-square solution of (36) for the true-solution case should have this property: Its error term is zero *theoretically* (or very close to zero due to rounding errors in computation). While the least-square solution for any non-solution case would have a *significant* (with respect to the rounding errors) nonzero error term, which makes the true solution stand out clearly. □

Our experiments have verified this phenomenon. Based on this observation, we develop a *two-stage least-square detection* method to minimize the computing cost, and in Section 5, we demonstrate the effectiveness of this method using a simple example.

**Remark 1.** *The overdetermined linear system (36) plays an important role in determining the true solution from certain number of possible cases. Typically, this situation happens in the multi-generator sparse expansion problem. For the single-generator case, we can select same number of linearly independent equations from the overdetermined system as the number of unknowns to find the solution; however, for the multi-generator case, the redundant equations are very useful in the least-square method.*

## 4. The Sparse Expansion Problem with Two Gaussian Generators

In this section, we solve another two-generator sparse expansion problem as in (7) that uses the two Gaussian generating functions, $e^{-\beta(\phi-x)^2}$ and $e^{-\beta(\phi+x)^2}$, in the form of

$$f(x) = \sum_{j=1}^{M} c_j e^{-\beta(\phi_j-x)^2} + \sum_{j=1}^{M} c_j e^{-\beta(\phi_j+x)^2} \tag{37}$$

for some constant $\beta \in \mathbb{C}\backslash\{0\}$. In order to recover the coefficients $c_j \in \mathbb{C}\backslash\{0\}$ and the parameters $\phi_j$'s, we need $4M$ sampling values $f(x_0 + kh), k = 0, \ldots, 4M - 1$, where $x_0 \in \mathbb{R}$, and $h$ satisfies the same condition as in Section 2.2.

This two-generator sparse expansion problem has a special property: When $\phi_{j_0} = 0$ for some $j_0 \in \{1, ..., M\}$, the two functions $e^{-\beta(\phi_{j_0}-x)^2}$ and $e^{-\beta(\phi_{j_0}+x)^2}$ are the same. This property would cause some problem for our method presented in the previous section. In order to make the discussion easier, we separate these two cases, and consider the case that $\phi_j \in \mathbb{R}\backslash\{0\}$ for all $j = 1, ..., M$ first.

**Theorem 2.** *Assume that a function $f(x)$ has the two-generator sparse expansion form of (37), where the number of terms $M$ and the constant $\beta \in \mathbb{C}\backslash\{0\}$ are known, but the coefficients in $\{c_1, \ldots, c_M\}$ and the parameters in $\{\phi_1, \ldots, \phi_M\}$ are unknown. If $4M$ equispaced sampling values of the form $f(x_0 + kh)$ for $k = 0, 1, \ldots, 4M - 1$ are provided, then the original function $f(x)$ can be uniquely reconstructed under the following conditions:*

> $1°$ *The coefficients $\{c_1, \ldots, c_M\}$ are nonzero in $\mathbb{C}$.*
> $2°$ *The parameters $\{\phi_1, \ldots, \phi_M\}$ are nonzero in $(-L, L) \subset \mathbb{R}$ for some $L > 0$, and they are distinct.*
> $3°$ *If $Re\,\beta \neq 0$, then $h \in \mathbb{R}\backslash\{0\}$; while if $Re\,\beta = 0$, then $0 < h \leq \frac{\pi}{2|Im\beta|L}$.*

**Proof.** Our method relies on existence of some *critical linear operator*, such that both generating functions are its eigenfunctions. Here we use the operator $\mathcal{S}_{K,h}$ as defined in (16) with $K(x, h) := e^{\beta h(2x+h)}$, which has the following properties:

$$(\mathcal{S}_{K,h}e^{-\beta(\phi-\cdot)^2})(x) = e^{2\beta\phi h}e^{-\beta(\phi-x)^2},$$
$$(\mathcal{S}_{K,h}e^{-\beta(\phi+\cdot)^2})(x) = e^{-2\beta\phi h}e^{-\beta(\phi+x)^2}. \tag{38}$$

Clearly $e^{-\beta(\phi_j-\cdot)^2}$ and $e^{-\beta(\phi_j+\cdot)^2}$ are eigenfunctions of $\mathcal{S}_{K,h}$ for all $\phi_j \in \mathbb{R}\backslash\{0\}$ with corresponding eigenvalues $e^{2\beta\phi_j h}$ and $e^{-2\beta\phi_j h}$, respectively, for $j = 1, ..., M$. Hence we can define the Prony polynomial using all these eigenvalues:

$$\Lambda(z) = \prod_{j=1}^{M}(z - e^{2h\beta\phi_j}) \prod_{j=1}^{M}(z - e^{-2h\beta\phi_j}) = \sum_{l=0}^{2M}\lambda_l z^l \tag{39}$$

with $\lambda_{2M} = 1$. Since the real number $\phi_j \neq 0$, we can assume that $\phi_j > 0$ for all $j = 1, ..., M$ based on the structure in (37) to improve the certainty without loss of generality.

Then for $m = 0, 1, ..., 2M - 1$, we calculate

$$\sum_{l=0}^{2M}\lambda_l(\mathcal{S}_{K,(l+m)h}f)(x_0) = \sum_{l=0}^{2M}\lambda_l e^{\beta h(l+m)(2x_0+h(l+m))}f(x_0 + h(l+m))$$

$$= \sum_{l=0}^{2M}\lambda_l e^{\beta h(l+m)(2x_0+h(l+m))}\sum_{j=1}^{M}c_j e^{-\beta(\phi_j-(x_0+h(l+m)))^2} + \sum_{l=0}^{2M}\lambda_l e^{\beta h(l+m)(2x_0+h(l+m))}\sum_{j=1}^{M}c_j e^{-\beta(\phi_j+(x_0+h(l+m)))^2}$$

$$= \left(\sum_{j=1}^{M}c_j e^{-\beta(x_0+hm-\phi_j)^2}e^{\beta hm(2x_0+hm)}\right)\underbrace{\left(\sum_{l=0}^{2M}\lambda_l e^{2\beta hl\phi_j}\right)}_{=0} + \left(\sum_{j=1}^{M}c_j e^{-\beta(x_0+hm+\phi_j)^2}e^{\beta hm(2x_0+hm)}\right)\underbrace{\left(\sum_{l=0}^{2M}\lambda_l e^{-2\beta hl\phi_j}\right)}_{=0} = 0,$$

which can be written as the following linear system

$$\sum_{l=0}^{2M-1}\lambda_l e^{\beta h(l+m)(2x_0+h(l+m))}f(x_0 + h(l+m)) = -e^{\beta h(m+2M)(2x_0+h(m+2M))}f(x_0 + h(m+2M)) \tag{40}$$

for $m = 0, 1, ..., 2M - 1$. To solve this system, we need $4M$ sampling values: $f(x_0 + kh)$ for $= 0, 1, \ldots, 4M - 1$. To study existence of the solution for this linear system, we would like to simplify it with respect to the unknown vector $\boldsymbol{\lambda} := [\lambda_0, ..., \lambda_{2M-1}]^T$ as follows,

$$H\boldsymbol{\lambda} = -G, \tag{41}$$

with $G := \left[ (\mathcal{S}_{K,(M+m)h} f)(x_0) \right]_{m=0}^{2M-1}$ and

$$H := \left[ (\mathcal{S}_{K,(l+m)h} f)(x_0) \right]_{l,m=0}^{2M-1}. \tag{42}$$

The invertibility of $H$ can be seen from the following matrix factorization:

$$
\begin{aligned}
H &= \left[ K(x_0, h(l+m)) f(x_0 + h(l+m)) \right]_{l,m=0}^{2M-1} \\
&= \left[ \sum_{j=1}^{M} c_j e^{\beta h(l+m)(2x_0 + h(l+m))} e^{-\beta(\phi_j - (x_0 + h(l+m)))^2} + \sum_{j=1}^{M} c_j e^{\beta h(l+m)(2x_0 + h(l+m))} e^{-\beta(\phi_j + (x_0 + h(l+m)))^2} \right]_{l,m=0}^{2M-1} \\
&= \left[ \sum_{j=1}^{M} c_j e^{-\beta(\phi_j - x_0)^2} e^{2\beta h(l+m)\phi_j} + \sum_{j=1}^{M} c_j e^{-\beta(\phi_j + x_0)^2} e^{-2\beta h(l+m)\phi_j} \right]_{l,m=0}^{2M-1} \\
&= V_h \operatorname{diag}\left( c_j e^{-\beta(\phi_j - x_0)^2} + c_j e^{-\beta(\phi_j + x_0)^2} \right) V_h^T \\
&= V_h D V_h^T
\end{aligned}
\tag{43}
$$

where the Vandermonde block matrix $V_h$ has the following form

$$V_h := \begin{bmatrix} A & | & B \end{bmatrix} \tag{44}$$

with

$$A := \begin{bmatrix} 1 & \cdots & 1 \\ e^{2\beta h \phi_1} & \cdots & e^{2\beta h \phi_M} \\ \vdots & \cdots & \vdots \\ e^{2(2M-1)\beta h \phi_1} & \cdots & e^{2(2M-1)\beta h \phi_M} \end{bmatrix}_{(2M) \times M} \tag{45}$$

and

$$B := \begin{bmatrix} 1 & \cdots & 1 \\ e^{-2\beta h \phi_1} & \cdots & e^{-2\beta h \phi_M} \\ \vdots & \cdots & \vdots \\ e^{-2(2M-1)\beta h \phi_1} & \cdots & e^{-2(2M-1)\beta h \phi_M} \end{bmatrix}_{(2M) \times M}, \tag{46}$$

and the diagonal block matrix $D$ is given by

$$D := \left[ \begin{array}{c|c} D1 & \mathbf{0} \\ \hline \mathbf{0} & D2 \end{array} \right] \tag{47}$$

with

$$D1 := \begin{bmatrix} c_1 e^{\beta(\phi_1 - x_0)^2} & & \\ & \ddots & \\ & & c_M e^{\beta(\phi_M - x_0)^2} \end{bmatrix} \tag{48}$$

and

$$
D2 := \begin{bmatrix} c_1 e^{\beta(\phi_1+x_0)^2} & & \\ & \ddots & \\ & & c_M e^{\beta(\phi_M+x_0)^2} \end{bmatrix}. \tag{49}
$$

From this structure, we can see that the Vandermonde matrix $V_h$ in (44) is invertible by conditions 2° and 3° of the theorem, and hence $H$ in (42) is also invertible by condition 1°, which results in the unique solution for $\lambda$.

With all the $\lambda_l$ values found from the above linear system, we can find all the $\phi_j$ values by calculating the zeros of the Prony polynomial of (39). In this case, we do not need to deal with the ambiguity that we encountered in the previous section due to the special structure of the pairs $(\phi_j, -\phi_j)$'s. Finally, the coefficients $c_j$'s of the sparse expansion (37) can be computed by solving the following *overdetermined* linear system:

$$
f(x_0 + lh) = \sum_{j=1}^{M} c_j \left( e^{-\beta(\phi_j+x_0-lh)^2} + e^{-\beta(\phi_j+x_0+lh)^2} \right) \tag{50}
$$

for $l = 0, ..., 4M - 1$. □

**Remark 2.** *Our method above only works for the case when $\phi_j \neq 0$ for all $j$ in $\{1, 2, \ldots, M\}$; however, in the real-world situation, when we solve a problem of (37) using 4M sampling values, how do we know if there exists any $\phi_j = 0$ in it or not? We need a detection method to tell us if all the $\phi_j$'s are nonzero before we apply the above method.*

Let us investigate the existence of a solution for the linear system (41), which is determined by the invertibility of $H$ in (42). We notice that when $\phi_1 = 0$, the first column of (45) and the first column of (46) are the same, which causes the matrix $V_h$ in (44) to be singular. Then, we conclude that $H$ in (43) is singular if any $\phi_j = 0$. In other words, by checking the invertibility of $H$, we can tell if there is any $\phi_j = 0$ for problem (37). If $H$ in (42) is singular, our current method does not work. Fortunately we can modify our method to solve the problem for this special situation.

Let us assume that $\phi_0 = 0$, and the remaining $\phi_j$'s are positive numbers. In this case, we modify (37) to

$$
f(x) = c_0 e^{-\beta x^2} + \sum_{j=1}^{M} c_j e^{-\beta(\phi_j-x)^2} + \sum_{j=1}^{M} c_j e^{-\beta(\phi_j+x)^2}, \tag{51}
$$

and its corresponding Prony polynomial is defined as

$$
\Lambda(z) = (z - 1) \prod_{j=1}^{M}(z - e^{2h\beta\phi_j}) \prod_{j=1}^{M}(z - e^{-2h\beta\phi_j}) = \sum_{l=0}^{2M+1} \lambda_l z^l \tag{52}
$$

with $\lambda_{2M+1} = 1$. Since $\Lambda(1) = 0$, it leads to

$$
\sum_{l=0}^{2M+1} \lambda_l = 0. \tag{53}
$$

Then we can show that

$$
\sum_{l=0}^{2M+1} \lambda_l (\mathcal{S}_{K,(l+m)h} f)(x_0) = 0, \quad \text{for } m = 0, 1, ..., 2M, \tag{54}
$$

because we can split the above left-hand-side summation into the following three summations with zero value each:

$$\sum_{l=0}^{2M+1} \lambda_l e^{\beta h(l+m)(2x_0+h(l+m))} c_0 e^{-\beta(x_0+h(l+m))^2} = c_0 e^{-\beta x_0^2} \underbrace{\sum_{l=0}^{2M+1} \lambda_l}_{=0} = 0,$$

$$\sum_{l=0}^{2M+1} \lambda_l e^{\beta h(l+m)(2x_0+h(l+m))} \sum_{j=1}^{M} c_j e^{-\beta(\phi_j-(x_0+h(l+m)))^2} = \left(\sum_{j=1}^{M} c_j e^{-\beta(x_0+hm-\phi_j)^2} e^{\beta hm(2x_0+hm)}\right) \underbrace{\left(\sum_{l=0}^{2M+1} \lambda_l e^{2\beta hl\phi_j}\right)}_{=0} = 0,$$

and

$$\sum_{l=0}^{2M+1} \lambda_l e^{\beta h(l+m)(2x_0+h(l+m))} \sum_{j=1}^{M} c_j e^{-\beta(\phi_j+(x_0+h(l+m)))^2} = \left(\sum_{j=1}^{M} c_j e^{-\beta(x_0+hm+\phi_j)^2} e^{\beta hm(2x_0+hm)}\right) \underbrace{\left(\sum_{l=0}^{2M+1} \lambda_l e^{-2\beta hl\phi_j}\right)}_{=0} = 0.$$

The linear system (54) for $\boldsymbol{\lambda} := [\lambda_0, ..., \lambda_{2M}]^T$ can be written as

$$\boldsymbol{H}\boldsymbol{\lambda} = -\boldsymbol{G}, \tag{55}$$

with $\boldsymbol{G} := \left[(\mathcal{S}_{K,(2M+m+1)h}f)(x_0)\right]_{m=0}^{2M}$ and

$$\boldsymbol{H} := \left[(\mathcal{S}_{K,(l+m)h}f)(x_0)\right]_{l,m=0}^{2M}. \tag{56}$$

We use $(4M+2)$ sampling values: $f(x_0 + kh)$ for $k = 0, 1, \ldots, 4M+1$ to solve the system. Similar to (43), we still have

$$\boldsymbol{H} = \boldsymbol{V}_h \boldsymbol{D} \boldsymbol{V}_h^T,$$

but we need to modify $\boldsymbol{V}_h$ to

$$\begin{bmatrix} 1 & 1 & \cdots & 1 & 1 & \cdots & 1 \\ 1 & e^{2\beta h\phi_1} & \cdots & e^{2\beta h\phi_M} & e^{-2\beta h\phi_1} & \cdots & e^{-2\beta h\phi_M} \\ \vdots & \vdots & \cdots & \vdots & \vdots & \cdots & \vdots \\ 1 & e^{4M\beta h\phi_1} & \cdots & e^{4M\beta h\phi_M} & e^{-4M\beta h\phi_1} & \cdots & e^{-4M\beta h\phi_M} \end{bmatrix}_{(2M+1)\times(2M+1)},$$

which is invertible for positive distinct $\{\phi_1, \ldots, \phi_M\} \subset (0, L)$, and the diagonal block matrix $\boldsymbol{D}$ becomes

$$\boldsymbol{D} = \begin{bmatrix} c_0 e^{-\beta x_0^2} & \boldsymbol{0} & \boldsymbol{0} \\ \hline \boldsymbol{0} & \boldsymbol{D1} & \boldsymbol{0} \\ \hline \boldsymbol{0} & \boldsymbol{0} & \boldsymbol{D2} \end{bmatrix}$$

with $\boldsymbol{D1}$ and $\boldsymbol{D2}$ maintaining the same forms of (48) and (49), respectively.

After we solve the linear system of (55), we obtain the Prony polynomial that contains one zero at $z = 1$ and the remaining zeros appear in pairs of $(z_j, z_j^{-1})$'s, which correspond to the parameter values 0 and $(\phi_j, -\phi_j)$ pairs. Finally, we will solve the following overdetermined linear system for $c_0, c_1, \ldots, c_M$ values

$$f(x_0 + lh) = c_0 e^{-\beta(x_0-lh)^2} + \sum_{j=1}^{M} c_j \left(e^{-\beta(\phi_j+x_0-lh)^2} + e^{-\beta(\phi_j+x_0+lh)^2}\right) \tag{57}$$

for $l = 0, ..., 4M+1$. From this example, we can see that the value of $\det(\boldsymbol{H})$ can give us some important information, that is, which of the two systems in (37) and (51) we should

work on. This property could be useful when we consider a problem in which the $M$ value in (51) is unknown, but restricted in certain range. (See discussion in Section 6).

## 5. Numerical Experiments

In this section, we use two simple examples to illustrate the implementation details of our method for the two-generator sparse expansion problem described in the previous sections. The first example is for version (23) in Section 3. The second example is for version (37) in Section 4.

**Example 1.** *We consider a function $f(x)$ (see Figure 1) that is a two-generator expansion with each generator producing* 5 *terms in the following form*

$$f(x) = \sum_{j=1}^{5} c_j \cos(\phi_j x) + \sum_{j=1}^{5} d_j \sin(\beta_j x), \tag{58}$$

*and the* 20 *parameters we used are listed in the table below to generate the sampling values.*

*How to use the* 39 *equispaced sampling values (where* 39 *comes from* $4(5+5)-1$) *in the form of* $f(x_0 + kh), k = -19, \ldots, 0, \ldots, 19$ *to recover the original parameters in Table 1?*

**Table 1.** Original parameters of the function $f(x)$ in (58).

| $j$ | $c_j$ | $d_j$ | $\phi_j$ | $\beta_j$ |
|---|---|---|---|---|
| 1 | $-2$ | 5 | 2 | 3 |
| 2 | 3 | $-6$ | 4 | 5 |
| 3 | $-4$ | 4 | 7 | 6 |
| 4 | 8 | $-3$ | 8 | 9 |
| 5 | 7 | 2 | 10 | 11 |

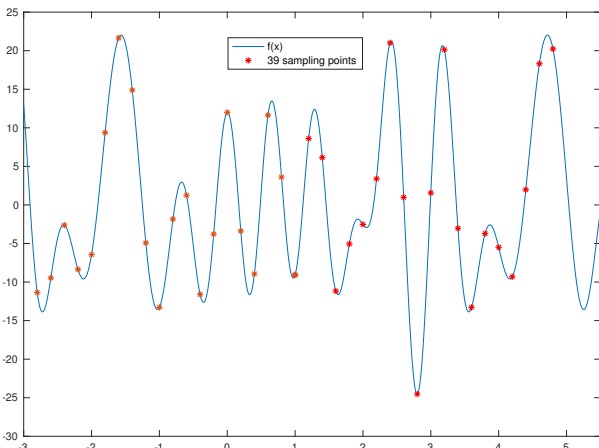

**Figure 1.** The signal $f(x)$ in (58) with 39 equispaced sampling values.

There are 20 original parameters in two sets: $\{c_1, \ldots, c_5, \phi_1, \ldots, \phi_5\}$ and $\{d_1, \ldots, d_5, \beta_1, \ldots, \beta_5\}$ corresponding to the two generators, respectively. To recover them, first we solve the following linear system for the coefficients of the Prony polynomial $\{\lambda_0, \ldots, \lambda_9\}$ based on the Equation (29)

$$H\lambda = -G,$$

where

$$H = \begin{bmatrix} -36.3064 & 24.4399 & 35.5543 & -40.2183 & -16.8633 & 18.9503 & -0.7573 & 18.8146 & 4.4668 & -52.7171 \\ 24.4399 & -0.3760 & -7.8892 & 9.3455 & -10.6340 & -8.8103 & 18.8824 & 1.8548 & -16.9513 & -9.4925 \\ 35.5543 & -7.8892 & -26.5849 & 21.6951 & 17.3985 & -10.7019 & -6.1982 & -16.8834 & -12.1046 & 24.6062 \\ -40.218 & 39.3455 & 21.6951 & -18.5319 & 21.6273 & 20.0106 & -46.4677 & -20.1576 & 24.6741 & 11.4779 \\ -16.8633 & -10.6340 & 17.3985 & 21.6273 & -15.9198 & -14.1386 & 6.0512 & -4.9102 & 3.4249 & 17.8445 \\ 18.9503 & -8.8103 & -10.7019 & 20.0106 & -14.1386 & -29.8792 & 27.4189 & 29.6337 & -11.7398 & -2.7658 \\ -0.7573 & 18.8824 & -6.1982 & -46.4677 & 6.0512 & 27.4189 & -6.2967 & 20.5893 & 23.4431 & -32.0445 \\ 18.8146 & 1.8548 & -16.8834 & -20.1576 & -4.9102 & 29.6337 & 20.5893 & -12.4873 & 0.2846 & 0.1035 \\ 4.4668 & -16.9513 & -12.1046 & 24.6741 & 3.4249 & -11.7398 & 23.4431 & 0.2846 & -35.8269 & 11.5787 \\ -52.7171 & -9.4925 & 24.6062 & 11.4779 & 17.8445 & -2.7658 & -32.0445 & 0.1035 & 11.5787 & -9.3042 \end{bmatrix}$$

and

$$G = \begin{bmatrix} 0.0458 & 0.0218 & -0.0275 & -0.0347 & -0.0103 & 0.0048 & 0.0510 & 0.0405 & -0.0520 & -0.0412 \end{bmatrix}^T.$$

We obtain

$$\lambda = \begin{bmatrix} -0.0088 & 0.0275 & -0.0639 & 0.1254 & -0.2113 & 0.3180 & -0.4300 & 0.5316 & -0.6010 & 0.3135 \end{bmatrix}^T,$$

which corresponds to the following Prony polynomial

$$\Lambda(z) = z^{10} + 0.3135z^9 - 0.6010z^8 + 0.5316z^7 - 0.4300z^6 + 0.3180z^5 - 0.2113z^4 + 0.1254z^3 - -0.0639z^2 + 0.0275z - 0.0088.$$

From the 10 zeros of this polynomial, we obtain 10 parameter values:

$$\{11.0000, 2.0000, 3.0000, 10.0000, 4.0000, 5.0000, 9.0000, 6.0000, 7.0000, 8.0000\}, \tag{59}$$

which correspond to $\{\phi_1, \ldots, \phi_5, \beta_1, \ldots, \beta_5\}$, but the explicit order is unknown. We must resolve the ambiguity: What five parameter values are for $\{\phi_1, \ldots, \phi_5\}$ (with the remaining five parameter values for $\{\beta_1, \ldots, \beta_5\}$)?

To separate the $\phi_j$'s from $\beta_l$'s, we consider the following overdetermined linear system:

$$\begin{bmatrix} \cos(\phi_1 x_0) & \cdots & \cos(\phi_5 x_0) & \sin(\beta_1 x_0) & \cdots & \sin(\beta_5 x_0) \\ \cos(\phi_1 x_1) & \cdots & \cos(\phi_5 x_1) & \sin(\beta_1 x_1) & \cdots & \sin(\beta_5 x_1) \\ \vdots & \vdots & \vdots & & \vdots & \vdots \\ \cos(\phi_1 x_{19}) & \cdots & \cos(\phi_5 x_{19}) & \sin(\beta_1 x_{19}) & \cdots & \sin(\beta_5 x_{19}) \end{bmatrix} \begin{bmatrix} c_1 \\ \vdots \\ c_5 \\ d_1 \\ \vdots \\ d_5 \end{bmatrix} = \begin{bmatrix} f_0 \\ f_1 \\ \vdots \\ f_{19} \end{bmatrix}, \tag{60}$$

where we use the shorthand notations

$$x_n = x_0 + nh \quad \text{and} \quad f_n = f(x_0 + nh)$$

for $n = 0, 1, \ldots, 19$. Note: In this linear system, we only use 20 out of 39 original sampling values, which is adequate for this particular example. It is a trade-off issue between the accuracy of computation and the cost of computation (in time). In general, the more redundant equations we use, the more accuracy we can achieve in searching for the true solution. In other words, if we can obtain adequate accuracy, we focus on cutting the computation cost to the minimum. We do not solve this overdetermined linear system by the *least-square* method directly. We split these 20 equations into two parts: In the first part, we approximate the coefficients $\{c_1, \ldots, c_5, d_1, \ldots, d_5\}$ in (60) by the least-square method. Then we apply these derived coefficients to the equations in the second part so as to filter out the true solution.

Among the 10 values in (59), every time we select 5 of them for $\{\phi_1, \ldots, \phi_5\}$, the remaining 5 numbers are automatically for $\{\beta_1, \ldots \beta_5\}$. We will have total 252 possible choices (which is the combinatorial number $\binom{10}{5}$) as the candidates for the solution. Notice that this combinatorial number is a relatively big number. In order to speed up the pro-

cessing, we reduce the redundant computation to the minimum. Let us use the notations $\{\phi_1^i, \ldots, \phi_5^i, \beta_1^i, \ldots \beta_5^i\}$ with $i = 1, 2, \ldots, 252$ representing those 252 candidates. Our method is based on the property that the information given in the sampling values has a lot of redundancy for selecting the true solution, and we only use just enough information from the given sampling values so as to save the computation time.

First, when we calculate the coefficients $\{c_1, \ldots, c_5, d_1, \ldots, d_5\}$ by the least-square method, we use exactly 10 equations (the same number of the coefficients) out of the 20 equations in (60). Based on our experiments, we do not have to use an overdetermined system for a good approximation by the least-square method. A determined system can give us excellent approximation for the least-square problem, while any underdetermined system usually does not approximate the data well through the least-square solution. For convenience, we select 10 consecutive equations in (60) somewhere in the middle, which we call the *least-square block* in our discussion, to approximate the coefficients $\{c_1, \ldots, c_5, d_1, \ldots, d_5\}$. Specifically, our least-square block takes the subscripts from 6 through 15, and the corresponding sampling values $\{f_6, f_7, \ldots, f_{15}\}$ should be selected as a reduced linear system given below,

$$
\begin{bmatrix}
\cos(\phi_1 x_6) & \cdots & \cos(\phi_5 x_6) & \sin(\beta_1 x_6) & \cdots & \sin(\beta_5 x_6) \\
\cos(\phi_1 x_7) & \cdots & \cos(\phi_5 x_7) & \sin(\beta_1 x_7) & \cdots & \sin(\beta_5 x_7) \\
\vdots & \vdots & \vdots & \vdots & \cdots & \vdots \\
\cos(\phi_1 x_{15}) & \cdots & \cos(\phi_5 x_{15}) & \sin(\beta_1 x_{15}) & \cdots & \sin(\beta_5 x_{15})
\end{bmatrix}
\begin{bmatrix} c_1 \\ \vdots \\ c_5 \\ d_1 \\ \vdots \\ d_5 \end{bmatrix}
=
\begin{bmatrix} f_6 \\ f_7 \\ \vdots \\ f_{15} \end{bmatrix}. \tag{61}
$$

Even if our new linear system (61) is a determined system, we still solve it for a least-square solution, because the determinant of the square matrix in (61) could be very close to zero. Then the remaining equations in (60) together with the coefficients derived from (61) will be used to detect which candidate is the true solution based on the error information.

For each set of values $\{\phi_1^i, \ldots, \phi_5^i, \beta_1^i, \ldots, \beta_5^i\}$ among the 252 candidates, the least-square solution for the linear system (61) would produce the 10 coefficients $[c_1^i, \ldots, c_5^i, d_1^i, \ldots, d_5^i]^T$, and we evaluate the following vector

$$
\begin{bmatrix} f_0^i \\ f_1^i \\ \vdots \\ f_{19}^i \end{bmatrix}
:=
\begin{bmatrix}
\cos(\phi_1^i x_0) & \cdots & \cos(\phi_5^i x_0) & \sin(\beta_1^i x_0) & \cdots & \sin(\beta_5^i x_0) \\
\cos(\phi_1^i x_1) & \cdots & \cos(\phi_5^i x_1) & \sin(\beta_1^i x_1) & \cdots & \sin(\beta_5^i x_1) \\
\vdots & \ddots & \vdots & \vdots & \ddots & \vdots \\
\cos(\phi_1^i x_{19}) & \cdots & \cos(\phi_5^i x_{19}) & \sin(\beta_1^i x_{19}) & \cdots & \sin(\beta_5^i x_{19})
\end{bmatrix}
\begin{bmatrix} c_1^i \\ \vdots \\ c_5^i \\ d_1^i \\ \vdots \\ d_5^i \end{bmatrix},
$$

which is in general different from the original sampling vector $[f_0, f_1, \ldots, f_{19}]^T$. Then we will calculate the difference of these two vectors, and see how close they are. We define the error vector as follows:

$$
\begin{bmatrix} \epsilon_0^i \\ \epsilon_1^i \\ \vdots \\ \epsilon_{19}^i \end{bmatrix}
:=
\begin{bmatrix} |f_0^i - f_0| \\ |f_1^i - f_1| \\ \vdots \\ |f_{19}^i - f_{19}| \end{bmatrix}. \tag{62}
$$

To search for the true solution among the 252 candidates, we discover an intrinsic property, shown in Figures 2 and 3, that can clearly separate the true solution from other candidates.

In Figure 2, we plot the error vector for one of the 252 candidates to view its typical behavior. The error values in the least-square block (with subscripts from 6 to 15) are very close to zero for a typical candidate; however, the error values that are out of the

least-square block (with subscripts from 0 to 5 and from 16 to 19) are not close to zero in general for a candidate that is not the true solution.

This behavior can be explained in this way: The errors in the least-square block are usually very small due to the fact that the least-square solution of the determined system approximates the targeting sampling values $\{f_6, f_7, \ldots, f_{15}\}$ quite well; however, when we consider an error for a sampling value out of the least-square block, since the corresponding equation is not involved in the least-square approximation, there is no reason for this equation to generate a value that is very close to the targeting sampling value.

While for the true solution case, the behavior is different in the sense that the errors for all the equations in the linear system (60) are very close to zero (see Figure 3, and ignore the two reference points at the ends). Let us summarize the key property that helps us to find out the true solution among all the candidates: *For a candidate, if the coefficients generated from the determined linear system (61) by the least-square method cannot approximate just one sampling value out of the least-square block well, then it cannot be the true solution.*

However, if the coefficients for one candidate can approximate one particular sampling value out of the least-square block well, we can only say that it is *highly likely* that this candidate could be the true solution, because the probability for a *non-solution* candidate to approximate some sampling value out of the least-square block well is very small. Based on this observation from our experiments, we design the following strategy for the solution search.

*Strategy*: Eliminate as many as possible candidates in the first round filtering in two steps: *Step 1.* Select a determined linear system from the overdetermined linear system in (60) (as the least-square block), and approximate the coefficients $\{c_1, \ldots, c_5, d_1, \ldots, d_5\}$ by the least-square method for each of the 252 candidates. *Step 2.* Apply the derived coefficients in *Step 1* on one of the linear equations out of the least-square block to approximate the targeting sampling value and calculate the error with the targeting sampling value. If the error is greater than certain threshold (we use 0.1 as our threshold), we drop this candidate from the consideration; otherwise, this candidate passes the first round filtering. If only one candidate survives the first round filtering, it must be the true solution. If more than one candidates pass the first round filtering, we need to do the second round filtering. In the second round filtering, we simply apply the derived coefficients on another linear equation out of the least-square block, and calculate the error for the targeting sampling value. If the error is greater than the threshold, we eliminate this candidate. We keep doing these cycles until we identify the true solution. Since we have plenty of redundant equations out of the least-square block, we should be able to determine the true solution without going through too many cycles in general. Furthermore, those linear equations corresponding to the original sampling values that are not included in the linear system (60) can still be used for the above steps when necessary, but the probability to use those equations out of the linear system (60) will be extremely small. This simple strategy is designed to allow us to detect the true solution without unnecessary computation, while we still preserve the option to use the redundant information when necessary.

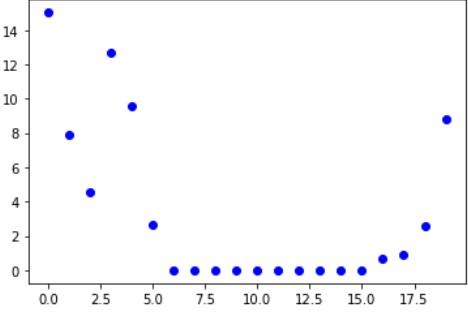

**Figure 2.** Display the error vector for one of the 252 candidates.

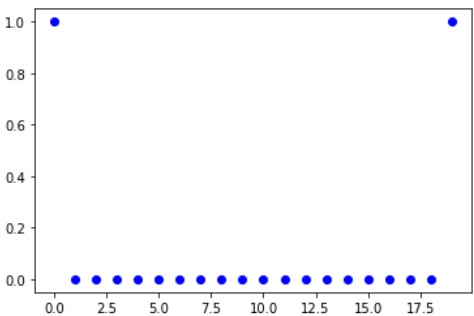

**Figure 3.** Display the error vector for the true solution with two reference points at the ends.

Here we would like to point out that as soon as we select values in $\phi$-group or $\beta$-group, the order of those values in each group is not important, because their corresponding coefficients ($c_j$'s or $d_l$'s) will also be aligned with them accordingly when we solve the determined linear system (61) using the least-square method.

**Example 2.** *Our second function to be recovered has the following form*

$$
\begin{aligned}
f(\omega) = {} & \frac{c_1}{2}\left(e^{-\frac{1}{2}(\phi_1-\omega)^2} + e^{-\frac{1}{2}(\phi_1+\omega)^2}\right) \\
& + \frac{c_2}{2}\left(e^{-\frac{1}{2}(\phi_2-\omega)^2} + e^{-\frac{1}{2}(\phi_2+\omega)^2}\right) \\
& + \frac{c_3}{2}\left(e^{-\frac{1}{2}(\phi_3-\omega)^2} + e^{-\frac{1}{2}(\phi_3+\omega)^2}\right),
\end{aligned}
\tag{63}
$$

*which is derived by applying the STFT on the following function*

$$
g(x) = \sum_{j=1}^{3} c_j \cos(\phi_j x),
\tag{64}
$$

*with the parameters of (64) listed in the following Table 2.*

**Table 2.** Parameters of the function $f(x)$ in (64).

| $j$ | $c_j$ | $\phi_j$ |
|---|---|---|
| 1 | 0.5000 | 1.0000 |
| 2 | 0.2500 | 3.0000 |
| 3 | 1.0000 | 4.0000 |

To solve this problem, we need to use 12 (i.e., $4M$) sampling values. After we applied the method described in Section 4, we solved a linear system with 6 unknowns, and derived the Prony polynomial of degree 6 as follows

$$
\Lambda(z) = 1.0000(z^6 + 1) - 14.4845(z^5 + z) + 65.9809(z^4 + z^2) + 108.8070z^3.
$$

The symmetric structure of this polynomial tells us that its zeros appear in $(z_j, z_j^{-1})$ pairs for $j = 1, 2, 3$, which correspond to three pairs of parameters: $(1.0000, -1.0000)$, $(3.0000, -3.0000)$, and $(4.0000, -4.0000)$ for $(\phi_j, -\phi_j), j = 1, 2, 3$. Finally, we can solve another linear system for the coefficients $c_j$'s with the errors listed in the Table 3.

**Table 3.** Parameters of the function $f(x)$ in (64) and approximate errors using 12 sampling values with $h = 0.5$.

| $j$ | $c_j$ | $\phi_j$ | $\|c_j - c_j^*\|$ | $\|\phi_j - \phi_j^*\|$ |
|---|---|---|---|---|
| 1 | 0.5000 | 1.0000 | $3.7970.10^{-2}$ | $5.2824.10^{-13}$ |
| 2 | 0.2500 | 3.0000 | $5.0987.10^{-14}$ | $4.5652.10^{-13}$ |
| 3 | 1.0000 | 4.0000 | $5.8065.10^{-14}$ | $1.4211.10^{-14}$ |

## 6. Conclusions

In this paper, we introduce a method that extends the Prony method to solve the *two-generator sparse expansion problem*. This method relies on the existence of a special linear operator for which the two generators must be the eigenfunctions. This two-generator problem has a special property: The zeros of its Prony polynomial correspond to two sets of parameters, and there is no straightforward way to separate them. We propose a *two-stage least-square detection method* on an overdetermined linear system for each candidate to extract the true solution, which relies on an intrinsic property for the true solution: *Only the true solution can use the coefficients derived from the least-square block to approximate the targeting sampling values out of the least-square block well.* Our method is designed to minimize the computation cost, while still maintain the computation accuracy.

It seems that the idea can be extended to the $k$-generator sparse expansion problem for $k > 2$; however, for the general $k$-generator case, the requirement that there exists a linear operator such that all the generators must be its eigenfunctions becomes *extremely* hard to achieve. For example, in the following sparse expansion problem,

$$f(x) = \sum_{j=1}^{M_1} c_j \cos(\phi_j x) + \sum_{l=1}^{M_2} d_l e^{\beta_l x}, \tag{65}$$

it is not easy to find a linear operator, such that both $\cos(\phi x)$ and $e^{\beta x}$ are its eigenfunctions. One may argue that the problem could be solved by converting $\cos(\phi x)$ to $\frac{1}{2}(e^{i\phi x} + e^{-i\phi x})$, and then it becomes a one-generator problem. Notice that converting a two-generator problem to a one-generator problem may not work most of the time. We are interested in developing a general method that can solve the two-generator sparse expansion problem including the one in (65). We can see that there are many difficult problems to be solved in this multi-generator sparse expansion problem, and we would like to see more researchers contribute in this direction.

Our method for the two-generator sparse expansion problem can handle certain degree of uncertainty. For example, in problem (23), if we know the total number of terms (i.e., the value of $M_1 + M_2$), but we do not know the number of terms in each summation (i.e., the individual values of $M_1$ and $M_2$), we can still solve the problem using our *two-stage least-square detection* method described in Sections 3 and 5. If we increase the uncertainty a little more, can we still solve the problem?

For example, in the problem we considered in Section 4, if we do not know the exact number of terms (it is referred to *unknown order of sparsity M* in [1]) in the following expansion,

$$f(x) = \sum_{j=1}^{M} c_j e^{-\beta(\phi_j - x)^2} + \sum_{j=1}^{M} c_j e^{-\beta(\phi_j + x)^2},$$

and we are given $K$ equispaced sampling values for some positive integer $K$. If we are told that these sampling values are sufficient to recover the signal, how do we recover it? In other words, we know that the number of terms $M$ is in the range $1 \leq M \leq \lfloor K/4 \rfloor$, but we do not know the exact number $M$, can we solve the problem? The answer is *yes*, because we can try all the possible cases: $M = 1, 2, \ldots, \lfloor K/4 \rfloor$, and for each case, we apply our *two-stage least-square detection* method to tell us if the true solution can be extracted.

However, we are not satisfied with this kind of *exhaustive search type* solution due to its high cost. We plan to develop an efficient *term number detection* method, so that when we make a term number prediction, this method can tell us if it is correct or not immediately. In [1], two methods are proposed: One is based on the rank of the **H** matrix, and the other is based on the singular values of the **H** matrix. The main issue is: How to obtain a *reliable* method to determine the $M$ value in the sparse expansion? Only after we obtain the correct term number we will pay the computation cost to go through all the necessary details to find the solution.

**Author Contributions:** Conceptualization, A.H. and W.H.; methodology, A.H. and W.H; software, A.H. and W.H.; validation, A.H. and W.H.; formal analysis, A.H. and W.H.; investigation, A.H. and W.H.; resources, A.H. and W.H.; data curation, A.H. and W.H.; writing—original draft preparation, A.H. and W.H.; writing—review and editing, A.H. and W.H.; visualization, A.H. and W.H. All authors have read and agreed to the published version of the manuscript.

**Funding:** This research received no external funding.

**Conflicts of Interest:** The authors declare no conflict of interest.

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
