# Peer review of "Prony Method for Two-Generator Sparse Expansion Problem"

_mca, doi:10.3390/mca27040060_

Round 1

Reviewer 1 Report

See enclosed review as pdf file.

Reviewer 2 Report

I reviewed this paper already last year for JCAM and asked for a major revision clearly indicating what is new beyond the more general approach by Plonka et al. - I never saw such a revision. Meanwhile, the authors incorporated a suggestion to replace a combinatorial step. Still the manuscript lacks a precise point what is new.

Author Response

Thank you for your helpful comments.

In Section 5, we created a new example based on your suggestion and found some interesting properties that were not available in our old example. Although the solution is much longer, it is much better than the old one. Your suggestion makes our paper better, and we appreciate it.

Round 2

Reviewer 1 Report

The manuscript is slightly improved. The results in sections 3 and 4 are now summarized in Theorems.

The authors should be more explicit  on page 3 (which range for h is appropriate ?) and on page 5 (which range for \phi_j do you mean?).

I do not agree with the arguments of the authors regarding their reply about feasibility of computations.  

First, one never would try to compute the inverse of a matrix in order to solve a linear system. This is content of Numerical Linear Algebra I.

Second,  a computational  cost of O(n^3) (polynomial cost) is not comparable with exponential cost that appears in case of the combinatorial search for the correct solution. 

The binomial (2n \choose n) is larger than 4^n/(2n+1), i.e., we have exponential growth!

I have however no other idea in this case, how to circumvent the combinatorial search. 

Therefore the paper can be published.

Author Response

Dear Reviewer, 
Thank again for your suggestions, we go through the whole paper carefully and try our best to fix all the possible problems. Please see the attached file. 

Best, 

Round 3

Reviewer 1 Report

The authors tried to improve the manuscript according to the review.